# Evaluation of the International Society for Cutaneous Lymphoma Algorithm for the Diagnosis of Early Mycosis Fungoides

**DOI:** 10.3390/cells10102758

**Published:** 2021-10-15

**Authors:** Hyang-Joo Ryu, Sun-Il Kim, Hyung-Ook Jang, Se-Hoon Kim, Sang-Ho Oh, Sujin Park, Sang-Kyum Kim

**Affiliations:** 1Department of Pathology, Yonsei University College of Medicine, Severance Hospital, Seoul 03722, Korea; hyangzo030@yuhs.ac (H.-J.R.); alexkim91@yuhs.ac (S.-I.K.); hoj3202@yuhs.ac (H.-O.J.); paxco@yuhs.ac (S.-H.K.); 2Department of Dermatology, Yonsei University College of Medicine, Severance Hospital, Seoul 03722, Korea; oddung93@yuhs.ac (S.-H.O.); parksj94@yuhs.ac (S.P.)

**Keywords:** mycosis fungoides, ISCL algorithm, CD5, CD7, TCR-γ, TCR-β

## Abstract

The International Society for Cutaneous Lymphoma (ISCL) proposes a diagnostic algorithm for early mycosis fungoides (MF) that includes clinical, histological, immunophenotypical, and molecular criteria. Here, we analyzed the immunologic markers and features of T-cell clonality in 38 early MF cases and 22 non-MF cases to validate the ISCL algorithm. We found that CD5 and CD7 expression differed significantly between early MF and non-MF cases, with epidermal discordance of CD7 expression more frequently identified in early MF. Notably, increasing the cut-off value for CD7 expression from 10% to 22.5% improved its sensitivity. Furthermore, TCR-γ and β chain rearrangements were more frequently detected in early MF than in non-MF cases. Based on these findings, we propose CD5 and CD7 deficiency as mandatory immunopathologic criteria and PCR-based testing for TCR-γ and β chains as required molecular/biologic criteria to improve the efficiency of early MF diagnosis using the ISCL algorithm.

## 1. Introduction

Mycosis fungoides (MF) is the most common type of primary cutaneous T-cell lymphoma and is characterized by the evolution of patches, plaques, and tumors on the skin [1]. Most cases of MFs wax and wane and exhibit indolent clinical courses; however, MF is incurable and can progress to the tumor stage with a poor prognosis [2,3]. Moreover, early MF at the patch stage has diverse clinicopathologic mimickers, from spongiotic or interface dermatoses to other cutaneous hematologic diseases [4], which can lead to misdiagnosis. Therefore, clinicians and pathologists must carefully evaluate clinicopathologic characteristics to accurately diagnose MF in its early stage.

The International Society for Cutaneous Lymphoma (ISCL) proposes an algorithm for the diagnosis of early MF and Sézary syndrome based on clinical, histologic, immunohistochemical (IHC), and molecular features, which are most important for recognizing classic MF at its initial presentation [5,6]. According to the ISCL algorithm, the diagnosis of early MF requires a total of four points for clinicopathologic and ancillary features. Clinical criteria are non-sun-exposed location, size/shape variation, or poikiloderma with persistent and/or progressive patches/thin plaques. Histopathological criteria are epidermotropism without spongiosis or lymphoid atypia with superficial lymphoid infiltrate. The molecular/biologic criterion is the presence of T-cell receptor (TCR) gene rearrangement based on polymerase chain reaction (PCR) testing. Lastly, immunopathologic criteria are CD2, CD3, CD5, and CD7 expression, although these ancillary features are not applicable if no clinical or histopathologic major criteria are fulfilled.

Several previous studies validated the ISCL diagnostic algorithm to demonstrate its effectiveness and accuracy [7,8]. These studies show some limitations of the algorithm, such as low specificity and accuracy of individual parameters, although the algorithm was able to detect significant differences between early MF and other mimickers. Although TCR gene rearrangement as a molecular/biologic criterion is included in the algorithm, details concerning TCR types or loci are not well described [5]. Among immunopathologic criteria, CD2, CD3, and/or CD5 deficiency and epidermal/dermal discordance of CD2, CD3, CD5, and/or CD7 have only ~10% sensitivity, whereas CD7 deficiency has ~40% sensitivity and 80% specificity [9,10,11,12]. Because the clinicopathologic features of early MF are often indistinguishable and complex, diagnosis is heavily dependent on ancillary features. Therefore, ascertaining reliable molecular and immunopathologic markers is needed to improve the diagnostic accuracy of the ISCL algorithm for early MF.

In this study, we classified cases into early MF or non-MF groups according to the ISCL diagnostic algorithm. By comparing each parameter of the algorithm between groups, we identified the most reliable biomarkers that can play a determinant role in efficiently diagnosing early MF and established new cut-off values that differentiate early MF from its mimickers.

## 2. Materials and Methods

### 2.1. Patients and Samples

We retrospectively obtained specimens from 60 patients who were biopsied with clinical suspicion of MF from 2008 to 2021 at Severance Hospital, Yonsei University College of Medicine, and reviewed their associated clinical presentation, microscopic findings, IHC staining results, and molecular studies. Slides stained with hematoxylin and eosin were reviewed by two pathologists (HJR and SKK). Ancillary studies were performed upon diagnosis. All methods and experimental protocols using human tissue were carried out in accordance with relevant guidelines and regulations as approved by the Institutional Review Board of Severance Hospital, Yonsei University Health System (IRB no. 4-2021-0462). The ISCL algorithm for the selection of eligible cases of early MF is shown in Figure 1. A total of 38 early MF and 22 non-MF specimens were included in this study.

The mimickers were clinically diagnosed as annular erythema, drug eruption, parapsoriasis, chronic eczema, pseudolymphoma, psoriasiform dermatitis, and lupus panniculitis after ICH and molecular work-up, although eight cases were still clinically suspicious for early MF (Appendix A).

### 2.2. IHC Staining

IHC staining was performed at diagnosis. Formalin-fixed paraffin-embedded (FFPE) blocks were cut into 4-μm-thick sections and processed using heat-induced epitope retrieval. IHC staining was performed in an automated immunostainer (Ventana Discovery^®^ XT, Ventana Medical System, Inc., Oro Valley, AZ, USA). Antibodies against CD2 (PA0271, Novocastra, 1:100, Buffalo Grove, IL, USA), CD3 (A0452, Dako, 1:100, Santa Clara, CA, USA), CD5 (NCL-CD5-4C7-L-CE, Novocastra, 1:25, Buffalo Grove, IL, USA), and CD7 (NCL-L-CD7-580, Novocastra, 1:100, Buffalo Grove, IL, USA) were utilized. Three pathologists (HJR, SHK, and SKK) blinded to the pathologic information independently evaluated IHC staining.

All immunohistochemical markers were assessed by light microscopy. The percentage of positive cells for immunostaining in the epidermis and dermis was measured, each percentage differed by at least 30%, epidermal discordance was defined [5,12]. If the expression of the markers were less than 30%, the complete loss of expression in the epidermis or the dermis was regarded as discordance.

### 2.3. DNA Extraction and TCR Gene Rearrangement Analysis

TCR gene rearrangement analysis was performed at diagnosis to evaluate TCR clonality. Genomic DNA was extracted from 10-μm sections cut from FFPE tissue blocks using the Maxwell^®^ CSC DNA FFPE extraction kit (Promega, Madison, WI, USA) and the Maxwell^®^ CSC instrument. Polymerization was performed with identiClone TCR-γ, β, and δ gene clonality assay kits according to the manufacturer’s instructions (Invivoscribe, San Diego, CA, US). After purification by electrophoresis and dying in 6% polyacrylamide for 2 h, analysis was conducted using the gel imaging system.

### 2.4. Statistical Analysis

All statistical analyses were performed using SPSS version 26.0 (IBM, Chicago, IL, USA) and GraphPad Prism version 7.04 (GraphPad Prism 7, San Diego, CA, USA). Categorical data are reported as frequencies and percentages, and continuous data are reported as mean and standard deviation (SD) or standard error of the mean (SEM). Differences between groups were compared using χ^2^ tests, Fisher’s exact tests, or unpaired t-tests. Receiver operator characteristic (ROC) curves were constructed for IHC results, and the area under the ROC curve (AUC) was calculated. To choose optimal cut-off values, Youden’s index [13] for each antibody was calculated, and Muller’s value was used as a measure of the discriminative ability of the AUC [14]. Two-sided *p*-values <0.05 were considered statistically significant. *P*-values are indicated as follows: * *p* < 0.05; ** *p* < 0.005; *** *p* < 0.0005.

## 3. Results

### 3.1. Clinicopathologic Features of Early MF

We reviewed 60 cases of pathologically diagnosed early MF (ISCL score ≥ 4, *n* = 38) and its mimickers (non-MF; ISCL score < 4, *n* = 22) according to the ISCL algorithm for the diagnosis of early MF and Sézary syndrome (Appendix A).

There was no significant difference in sex or age at diagnosis between groups. As expected, however, there were significant differences between groups with respect to clinical, histopathologic, molecular/biologic, and immunopathologic parameters (Table 1).

### 3.2. Determinative Immunopathologic Markers of Early MF

Figure 2 shows representative clinical features of early MF during the patch stage, including the histologic findings of epidermotropism and lymphoid atypia and the immunopathologic features of CD2, CD3, CD5, and CD7 deficiency (Figure 2a).

Next, we compared the expression of each immunopathologic marker between early MF and non-MF groups to identify the most reliable markers and their cut-off values (Table 2 and Figure 2b).

We found that CD5 expression was significantly lower in early MF (81.18 ± 3.438%) than in non-MF (92.50 ± 1.175%, *p* = 0.0174), although there was no significant difference in epidermal discordance in CD5 expression by dermal T-cells. CD7 expression was also significantly lower in early MF (17.45 ± 2.648%) than in non-MF (42.95 ± 4.763, *p* < 0.0001), and epidermal discordance in CD7 expression by dermal T-cells was more frequently identified in early MF (15/38, 39.5%) than in non-MF (0/22, 0%, *p* = 0.0010). There were no significant differences in CD2 and CD3 expression between groups.

Next, we constructed ROC curves and calculated AUCs for immunopathologic markers (Figure 2c and Table 3). Youden’s index was calculated for each marker to determine its optimal cut-off value (Appendix A).

According to Muller’s value, the AUCs of CD2 and CD3 showed poor and failing discriminatory abilities, respectively, at the highest Youden’s index. CD2 had a high cut-off value (92.5%) with low sensitivity (60.5%) and specificity (68.2%), although it was statistically significant (*p* = 0.031). CD3 also had a high cut-off value (92.5%) with very low sensitivity (28.9%, *p* = 0.696). CD5 and CD7 showed fair and good discriminatory abilities, respectively. CD5 had a high cut-off value (92.5%) with 65.8% sensitivity and 72.7% specificity (*p* = 0.005). CD7 had a cut-off value of 17.5% at the highest Youden’s index with 55.3% sensitivity and 100.0% specificity. However, if we chose 22.5% as the cut-off value for CD7, it had 73.7% sensitivity and 77.3% specificity (*p* < 0.001).

### 3.3. Molecular Features of Early MF

To specify the features of clonal TCR gene rearrangement in early MF, we performed PCR-based detection of TCR-γ, δ, and β chain genes (Figure 3a). We found that 20 out of 60 cases had oligoclonal TCR rearrangements (33.3%). Whereas 36 out of 38 early MF cases (94.7%) had monoclonal or oligoclonal T-cells, only 3 out of 22 non-MF cases (9.1%) had monoclonal or oligoclonal T-cells (*p* < 0.0001). TCR-γ and β gene rearrangements were more frequently identified in early MF than in non-MF (Table 4 and Figure 3b). TCR-β gene rearrangement was not detected in non-MF cases. There was no difference in TCR-δ gene rearrangement between groups (*p* > 0.9999).

## 4. Discussion

We classified early MF and non-MF cases according to the ISCL diagnostic algorithm and analyzed each parameter in the algorithm to improve the accuracy and reliability of ancillary findings. We found that total, clinical, pathologic, molecular, and immunopathologic scores differed significantly between groups. Therefore, benign mimickers should easily be able to be excluded according to the ISCL diagnostic algorithm. However, a diagnostic problem still remains because the clinical features of early MF vary greatly across patients [4,15], and the histologic features of MF during the patch stage are subtle, as lymphocytic infiltrates can be scant and lymphocytic atypia can be minimal [16,17]. If clinico-histologic features are similar and the sum of clinical and histologic scores is <4 according to the ISCL algorithm, immunopathologic and molecular/biologic features can play a determinant role in diagnosing early MF.

Although the ISCL algorithm clearly specifies the antibodies that must be stained, their cut-off values are undependable due to their low sensitivities [9,10,11,12]. Thus, we calculated the cut-off value of each marker based on its ROC curve to improve the reliability of molecular/biologic criteria. We found that CD2 and CD3 expression were similar between groups and had poor and failing discriminatory abilities, respectively. The optimal cut-off value for CD2, CD3, and CD5 expression was 92.5%, although the ISCL algorithm recommends a cut-off value of <50% for each marker [5,6]. Actually, only three out of 38 early MF cases showed less than 50% expression of CD2, CD3, or CD5 in this study. Furthermore, a 92.5% cut-off value is so high that it is difficult to differentiate the loss of expression from a reactive pattern. CD5 expression differed significantly between groups, but there were only two cases of early MF with CD5 epidermal discordance. Notably, however, CD7 expression and its epidermal discordance not only differed significantly between groups but also had a good discriminatory ability, with 73.7% sensitivity and 77.3% specificity at a 22.5% cut-off value. Although the ISCL diagnostic algorithm recommends a <10% cut-off value for CD7 expression, we found that only 18 out of 38 early MF cases (47.4%) showed <10% CD7 expression. Instead, we identified 22.5% as an optimal cut-off value for CD7 expression, with 28 out of 38 early MF cases (73.7%) showing <22.5% CD7 expression. Therefore, we propose that evaluation of CD5 and CD7 expression should be included in the diagnostic criteria for early MF, but CD2 and CD3 expression are not helpful for diagnosis. When the new cut-off values of CD5 and CD7 expression are applied to this data set, three non-MF cases are re-diagnosed as early MF and then, the specificity of the ISCL diagnostic algorithm could rise from 63.6% to 73.7%.

The ISCL diagnostic algorithm for early MF clearly documents the methodology for PCR-based analysis of TCR gene rearrangement. However, critical clones and features of TCR genes are not included. To evaluate these molecular features, we performed PCR-based testing to detect the monoclonality or oligoclonality of TCR-γ, δ, and β chains. We found that TCR-γ and β chains were frequently mutated in early MF but that there was no difference in TCR-δ mutation between early MF and non-MF groups. Notably, TCR-β clonality was not identified in any non-MF cases, therefore we believe that TCR-β clonality has 100% specificity in early MF. TCR rearrangements were also detected in 4 out of 22 non-MF cases in this study, which are presumed to be persistent cutaneous inflammatory infiltrates of monoclonal or oligoclonal T-cells due to conditions secondary to autoimmune or iatrogenic immune dysregulation and chronic idiopathic dermatoses [18,19,20].

In this study, after making pathologic diagnoses, dermatologists changed the clinical diagnosis from mycosis fungoides to parapsoriasis in 7 out of 22 non-MF cases. Parapsoriasis describes a group of cutaneous diseases that can be characterized by scaly patches or slightly elevated papules and/or plaques dispersed on the trunk or proximal extremities that have a resemblance to psoriasis [21]. Histologically, it shows cutaneous lymphoproliferations and refers to a heterogeneous group of uncommon dermatoses similar to psoriasis. It is broadly divided into two main types: small plaque parapsoriasis (SPP) and large plaque parapsoriasis (LPP). While SPP is generally considered a chronic benign condition, LPP is regarded as a premalignant dermatosis with a substantial risk of progression to mycosis fungoides. Actually, 10% to 30% of LPP cases progress to a frank cutaneous T-cell lymphoma [22]. It is known that monoclonal populations of T-cells could be found in 20% or more cases of LPP [23]. Therefore, the major differential diagnosis of parapsoriasis includes early-stage MF and chronic eczema, which usually cannot be distinguished on histologic grounds alone [24]. Thus, we included parapsoriasis as one of the mimickers of mycosis fungoides. However, there is considerable debate on the terminology and relation of parapsoriasis to MF. We believe that there was still a possibility of the evolution of mycosis fungoides in parapsoriasis cases involved in this study, and parapsoriasis could not be entirely defined by the ISCL algorithm.

In conclusion, we propose CD5 and CD7 deficiency as mandatory immunopathologic criteria and PCR-based testing for TCR-γ and β chains as required molecular/biologic criteria to improve the efficiency of diagnosing early MF using the ISCL algorithm. In particular, the cut-off value of CD7 expression should be increased from <10% to <22.5% to improve the discriminatory ability of this criterion.

## Figures and Tables

**Figure 1 cells-10-02758-f001:**
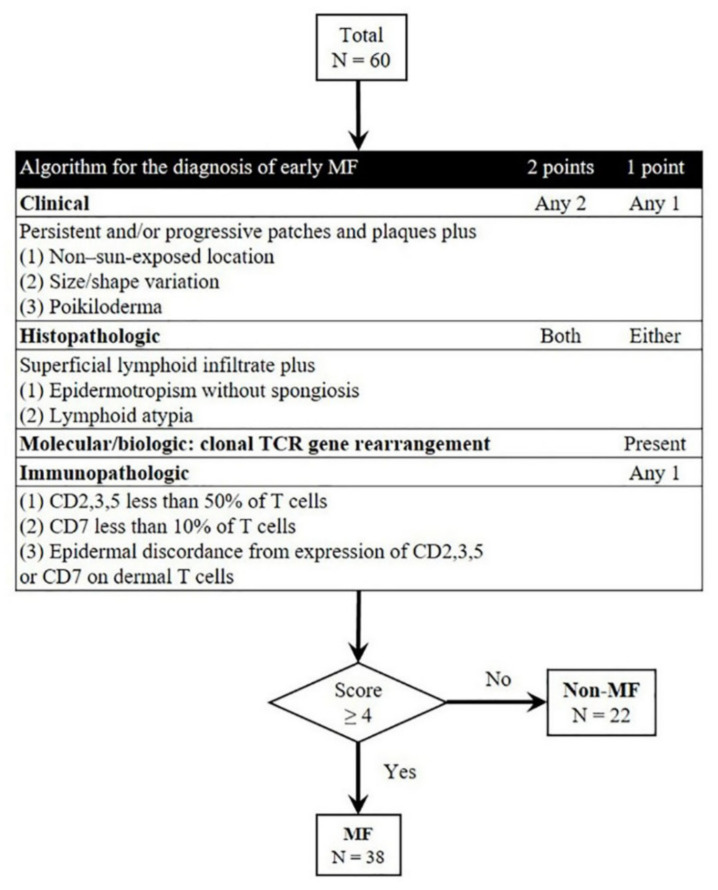
ISCL algorithm for the selection of early MF cases.

**Figure 2 cells-10-02758-f002:**
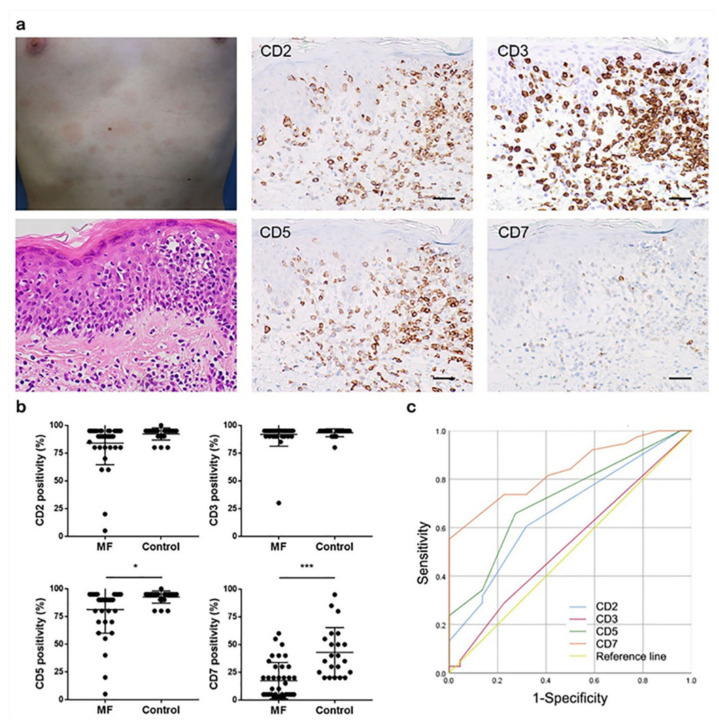
Clinicopathologic features of early MF. (**a**) Representative clinicopathologic findings during the patch stage of MF. Magnification, 200×; scale bar, 50 μm. (**b**) Comparative analyses of CD2, CD3, CD5, and CD7 expression. (**c**) ROC curves for CD2, CD3, CD5, and CD7 expression. * *p* < 0.05; *** *p* < 0.0005.

**Figure 3 cells-10-02758-f003:**
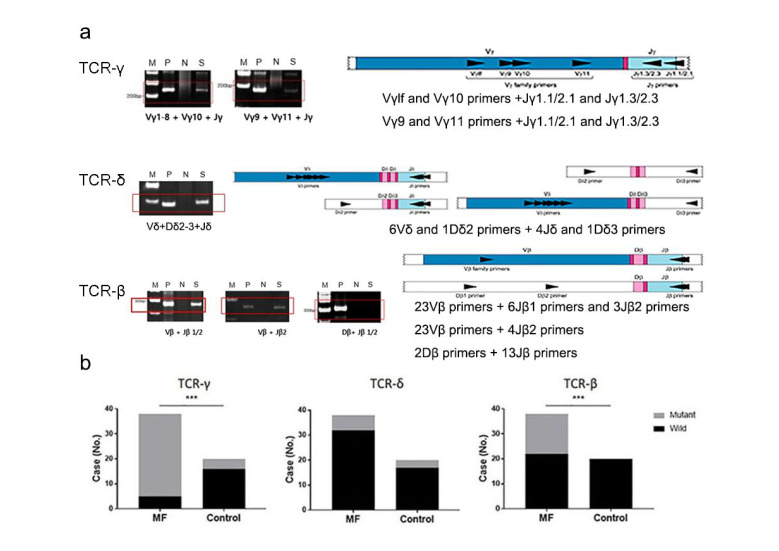
Clonal TCR gene rearrangement in early MF and non-MF. (**a**) PCR-based testing results. (**b**) Quantification of TCR gene rearrangement. *** *p* < 0.0005.

**Table 1 cells-10-02758-t001:** Clinicopathologic features of early MF and non-MF.

	Early MF (*n* = 38)	Non-MF (*n* = 22)	*p*-Value
Sex (male:female)	22:16	10:12	0.4254
Age (years, mean ± SD)	43.13 ± 2.858	51.91 ± 3.896	0.0717
Total ISCL score (mean ± SD)	4.868 ± 0.1261	2.0 ± 0.1861	<0.0001
Clinical points			<0.0001
0 1 2	1 7 30	9 8 5	
Histopathologic points			0.0500
0 1 2	3 15 20	5 12 5	
Molecular/biologic points			<0.0001
0 1	2 34	16 4	
Immunopathologic points			<0.0001
0 1	11 25	20 0	

**Table 2 cells-10-02758-t002:** Immunopathologic features of early MF and non-MF.

	Early MF (*n* = 38)	Non-MF (*n* = 22)	*p*-Value
IHC (%, ± SEM) CD2 CD3 CD5 CD7			
	84.08 ± 3.155 91.84 ± 1.722 81.18 ± 3.438 17.45 ± 2.648	92.27 ± 1.174 93.41 ± 0.764 92.50 ± 1.175 42.95 ± 4.763	0.0590 0.5065 0.0174 <0.0001
Epidermal discordance CD2 CD3 CD5 CD7	0 0 2 15	0 0 0 0	>0.9999 >0.9999 0.5402 0.0010

**Table 3 cells-10-02758-t003:** Data from ROC curves for CD2, CD3, CD5, and CD7 expression in early MF versus non-MF.

IHC	AUC	Muller’s Value	SE	95% Confidence Interval		*p*-Value	Cut-Off (%)	Sensitivity (%)	Specificity (%)
				Lower	Upper				
CD2 CD3 CD5 CD7	0.669 0.531 0.718 0.837	Poor Fail Fair Good	0.071 0.077 0.067 0.050	0.529 0.379 0.587 0.739	0.808 0.682 0.849 0.934	0.031 0.696 0.005 0.000	92.5 92.5 92.5 22.5	60.5 28.9 65.8 73.7	68.2 77.3 72.7 77.3

**Table 4 cells-10-02758-t004:** Molecular profiles of early MF and non-MF.

	Early MF (*n* = 38)	Non-MF (*n* = 22)	*p*-Value
TCR gene rearrangement TCR-γ TCR-δ TCR-β			<0.0001
	33 (86.8%) 6 (15.8%) 16 (42.1%)	4 (18.2%) 3 (13.6%) 0 (0.0%)	<0.0001 >0.9999 0.0002

## Data Availability

Data are available from the corresponding author upon request.

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
