# Peer review of "Evaluation of the International Society for Cutaneous Lymphoma Algorithm for the Diagnosis of Early Mycosis Fungoides"

_cells, 2021, doi:10.3390/cells10102758_

Round 1

Reviewer 1 Report

This manuscript analyses the validity of the ISCL algorithm for the diagnosis of early MF on an institutional cohort by comparing MF cases with potential mimickers. While the manuscript has its merits, the inclusion of patients with "parapsoriasis" in its control group creates concerns as this controversial entity is regarded by most authorities in the field of cutaneous lymphoma as early non-diagnostic forms of MF and thus represents an inclusion bias. Likewise the inclusion of patients with "lupus panniculitis" amongst controls is not adequate as this entity has a lymphocytic infiltrate centered on the subcutis (instead of being a superficial dermal infiltrate with epidermotropism) and is not a real simulator of MF neither clinically nor histopathologically. Finally the number of controls is smaller than that of cases classified as MF which is also concerning. 

Author Response

This study was conducted on patients who were clinically suspected of mycosis fungoides and who had undergone biopsy. The ancillary studies such as immunohistochemistry and gene rearrangement analysis were performed at diagnsoses as we mentioned in the Materials and Methods section. We retrospectively reviewed about 120 cases and excluded cases which were diagnosed as other hematologic malignancies after ancillary studies or some of the ancillary studies were not conducted at diagnosis. Finally, we included a total of 38 early MF and 22 non-MF specimens. As a retrospective study, not a prospective study, the small number of control group could be a limitation of this study.

Parapsoriasis describes a group of cutaneous diseases that can be characterized by scaly patches or slightly elevated papules and/or plaques dispersed on the trunk or proximal extremities that have a resemblance to psoriasis. Histologically, it shows cutaneous lymphoproliferations and refers to a heterogeneous group of uncommon dermatoses smilar to psoriasis. It is broadly divided in two main types: small plaque parapsoriasis (SPP) and large plaque parapsoriasis (LPP). While SPP is generally considered a chronic benign condition, LPP is regarded as a premalignant dermatosis with a substantial risk of progression to mycosis fungoides. The lymphocytic infiltrate of LPP is composed of small lymphocytes, some of which may have cerebriform, convoluted nuclei. There is focal lymphocytic epidermotropism, but Pautrier microabscesses, a classic histologic characteristic of mycosis fungoides, are usually absent in LPP. Therefore, we included parapsoriasis as one of the mimickers of mycosis fungoides. In this study, after making pathologic diagnoses, dermatologists changed the clinical diagnosis from mycosis fungoides to parapsoriasis in 7 out of 22 non-MF cases. We believe that the ISCL diagnostic algorithm is useful for differentiating parapsoriasis from early mycosis fungoides.

On out of 22 non-MF cases was finally diagnosed as lupus panniculitis by the dermatologists after ICH and molecular work-up. Lupus panniculitis is an autoimmune lobular lymphocytic panniculitis which shares features with subcutaneous panniculitis-like T cell lymphoma. When we received the specimen for the pathologic diagnosis, it did not contain the subcutaneous fat tissue, and we could find out the lymphocytic infiltration in the upper dermis and performed IHC and molecular tests according to the ISCL algorithm. After the pathologic diagnosis, re-biopsy was performed in the deeper under the skin and it was regarded as lobular or mixed lobular and septal panniculitis. Finally, the patient was clinically diagnosed as lupus panniculitis.

We added the changed clinical diagnosis in the Supplementary table 1 of the revised manuscript. Also we corrected the sentences as below in the Materials and Methods section (page 3):

The mimickers were clinically diagnosed as annular erythema, drug eruption, parapsoriasis, chornic eczema, pseudolymphoma, psoriasiform dermatitis, and lupus panniculitis after ICH and molecular work-up, although eight cases were still clinically suspicious for early MF (Supplementary Table 1).

Reviewer 2 Report

Comments:

Overall, a very well-written paper. The subject is highly relevant to the field. 

Critiques:

1) one of the major challenges in making histologic assessment of MF cases is related to the inter-observer variability. Based on the described methods, there were 3 pathologists who performed the IHC (and presumably histology) assessment. It will be ideal if the authors can present the variability among the three pathologists regarding the presence/absence of epidermotropism without spongosis and cytologic atypia. Similarly, how consistent the pathologists were in quantifying the loss of cd7? Based on Table 2, the error (standard deviation) related to the assessment of loss of CD7 is 2.6%, which is rather 'too good to be true' for three observers. The authors need to better elaborate on how they gathered the data

2) Of the 38 MF cases (out of 60) that have a ISCL score of 4 or higher,  how many of these patients clinically behave like MF? some followup clinical info will be ideal. Do any of these MF patients demonstrate 'atypical' clinical features, which may cast doubts to the diagnosis. Conversely, how confident the authors are regarding the diagnoses for the non-MF group. In other words, the ISCL algorithm (as stated by the authors) is not perfect, and the authors need to present more data to support their diagnoses used in this study.

3) one of the criteria for IHC is related to 'discordant expression of CD2, CD5 and CD7. This criterion is rather subjective. Have the authors done work in better refining this criterion, so that it can be used in a more reliable and consistent manner?

4) lastly, since TCR-beta is 100% specific - shouldn't this marker deserve more weight? Have the authors tried to adjust how the scores were calculated, based on their findings? Regarding the high specificity of TCR-beta in diagnosing MF, can the author include the data of some previously published works for comparison. In my own experience, TCR-beta is 100% specific. 

Minor

1) line 199 and 201 - I do not think the use of 'mutation' is appropriate for TCR. This is a test for monoclonality, not gene mutation per se

Author Response

  1. One of the major challenges in making histologic assessment of MF cases is related to the inter-observer variability. Based on the described methods, there were 3 pathologists who performed the IHC (and presumably histology) assessment. It will be ideal if the authors can present the variability among the three pathologists regarding the presence/absence of epidermotropism without spongosis and cytologic atypia. Similarly, how consistent the pathologists were in quantifying the loss of cd7? Based on Table 2, the error (standard deviation) related to the assessment of loss of CD7 is 2.6%, which is rather 'too good to be true' for three observers. The authors need to better elaborate on how they gathered the data

Reply:

Three well trained pathologists (HJR, SHK, and SKK) are dermatopathologists who had the experiences about making diagnoses of skin biopsy tissues over two years. The presence or absence of epidermotropism was separately evaluated on the same glass slide to get rid of heterogeneity of the lesion which could be varied by cutting.

To evaluate the CD2, CD3, CD5, and/or CD7 deficiency and epidermal/dermal discordance, we counted marker-positive cells and total lymphocytes in the epidermis and the dermis as the cited reference [#12, page 9] which was listed in the submitted original manuscript (Am J Pathol . 1990 Dec;137(6):1447-51). Most early MF cases (35 out of 38 cases) showed epidermotropism. Especially, reduced CD7 expression was more frequently identified, ranged from 3% to 60% (the mean expression was 17.45%). Only a few cases (4 out of 38 cases) showed over 50% expression and the most cases showed CD7 expression between 5% and 20% (see below the graph).

We performed unpaired-t test for the comparison between MF and non-MF groups, and actually, we calculated standard error for the mean (SEM), not standard deviation (SD). We added IHC (%,± SEM) in the Table 2 of the revised manuscript to clarify the values.

  1. Of the 38 MF cases (out of 60) that have a ISCL score of 4 or higher, how many of these patients clinically behave like MF? some followup clinical info will be ideal. Do any of these MF patients demonstrate 'atypical' clinical features, which may cast doubts to the diagnosis. Conversely, how confident the authors are regarding the diagnoses for the non-MF group. In other words, the ISCL algorithm (as stated by the authors) is not perfect, and the authors need to present more data to support their diagnoses used in this study.

Reply:

             38 cases were clinicopathologically confirmed the diagnosis as early MF according to the ISCL algorithm. However, the mimickers were clinically diagnosed as annular erythema, drug eruption, parapsoriasis, chornic eczema, pseudolymphoma, psoriasiform dermatitis, and lupus panniculitis after ICH and molecular work-up, although eight cases were still clinically suspicious for early MF (Supplementary Table 1). Therefore, we can calculate the sensitivity and the specificity of the ISCL algorithm, 100.0% and 63.6%, respectively. We believe that the ISCL diagnostic algorithm has high sensitivity but low specificity. In this study, we presented new cut-off values for each markers of the immunopathologic criteria. When the new cut-off values are applied to this data set, the specificity could rise upto73.7%. We added this in the discussion section of the revised manuscript as below (page 7):

             When the new cut-off values of CD5 and CD7 expression are applied to this data set, three non-MF cases re-diagnosed as early MF and then, the specificity of the ISCL di-agnostic algorithm could rise from 63.6% to 73.7%.

  1. One of the criteria for IHC is related to 'discordant expression of CD2, CD5 and CD7. This criterion is rather subjective. Have the authors done work in better refining this criterion, so that it can be used in a more reliable and consistent manner?

Reply: To address your comment, we added more details about immunohistochemical staining interpretation in “Materials and Methods” section (page 3), as below:

All immunohistochemical markers were assessed by light microscopy. The percent-age of positive cells for immunostaining in the epidermis and dermis was measured, each percentage differed by at least 30%, epidermal discordance was defined as the ISCL algorithm [5, 12]. If the expression of the markers were less than 30%, the complete loss of expression in the epidermis or the dermis was regarded as discordance.

  1. Lastly, since TCR-beta is 100% specific - shouldn't this marker deserve more weight? Have the authors tried to adjust how the scores were calculated, based on their findings? Regarding the high specificity of TCR-beta in diagnosing MF, can the author include the data of some previously published works for comparison. In my own experience, TCR-beta is 100% specific.

Reply: According to the reviewer’s comment, we added the details about the high specificity of TCR-β mutation in early MF as below (page 8):

             Notably, TCR-β clonlity was not identified in any non-MF cases, therefore we believe that TCR-β clonality has 100% specificity in early MF.

Minor

1) line 199 and 201 - I do not think the use of 'mutation' is appropriate for TCR. This is a test for monoclonality, not gene mutation per se

Reply: To address your comment, we edited the term of ‘mutation’ to ‘clonality’ so that we could explain the detection of monoclonality or oligoclonality of TCR genes.

Round 2

Reviewer 1 Report

I have reviewed the authors replies and unfortunately I still believe that there are too many biases for this paper to be acceptable for publication 

It is the opinion of the referee that the paper herein is not suitable for publication. As previously stated in the first review of the submitted manuscript, while the paper has its merits, the inclusion of patients with "parapsoriasis" in its control group creates major concerns as this controversial entity is regarded by most authorities in the field of cutaneous lymphoma as early non-diagnostic forms of MF, and thus represents a serious inclusion bias in the present study. Particularly since 32% of the control group herein represents cases labeled as parapsoriasis and given that 8 out of 22 theoretical ¨non-MF¨ cases were still deemed suspicious for MF based on clinical features. A proper study to validate the ISCL algorithm for the diagnosis of early MF should only include amongst its controls bona fide reactive dermatoses (true inflammatory mimickers) such as psoriasis, eczema, drug reactions, etc. and not ¨controversial¨ dermatoses that could represent early non-diagnostic forms of MF, such as parapsoriasis.  Parapsoriasis is a problematic term—the accuracy of which has been questioned since it was first named at the beginning of the 20th century. Since then, it has been considered a subject without a clear elucidation of its real nature. The name was coined because of clinical similarities with psoriasis, but throughout its history parapsoriasis has been considered a misnomer. In the early 1990s, Ackerman proposed that parapsoriasis guttata and digitate dermatoses could be considered two of many MF faces. If simultaneous biopsies were taken from distinct sites, it was questionable whether all criteria for MF will appear in only one biopsy. The other samples could show such subtle criteria that pathologists may not be comfortable making a diagnosis. Clinically, there is a temporal gap between the clinical presentation and the histopathological features of MF, which varies among patients. There have been no clinical or histopathological signs supporting the diagnosis of parapsoriasis or even predicting the risk of progression of so-called parapsoriasis lesions to lymphoproliferative disorders. According to Jaffe et al, parapsoriasis is a scientifically invalid term that could be understood as clinician's doubt if the lesion is in fact a patch phase of MF. Parapsoriasis in summary remains an ill-defined term, showing poorly understood nosology. Considering the modern dermatopathology practice based on reproducible strict criteria, diverse authors have suggested that the term “parapsoriasis” should be excluded from medical vocabulary. In the absence of an adequate term, in those cases when there is a suspicion of MF and the clinical and/or histopathological findings are quite subtle for a final diagnosis, descriptive reports and close follow-up are the most suitable approach. Furthermore, differentiating so called parapsoriasis from early-stage MF by the presence or absence of Pautrier microabscesses is erroneous since these are absent in a large proportion of cases of early-patch stage MF, thus making it imperative for the diagnosis to be rendered upon careful clinico-pathologic correlation and longitudinal follow up. The following references may serve to further comprehend the importance of not including cases of so called parapsoriasis in a study such as the present one:

King-Ismael D, Ackerman AB. Guttate parapsoriasis/digitate dermatosis (small plaque parapsoriasis) is mycosis fungoides. Am J Dermatopathol. 1992;14(6):518-530.

Fleischmajer R, Pascher F, Sims CF. Parapsoriasis en plaques and mycosis fungoides. Dermatologica. 1965;131(3):149-160

Lazar AP, Caro WA, Roenigk HH Jr, Pinski KS. Parapsoriasis and mycosis fungoides: the Northwestern University experience, 1970 to 1985. J Am Acad Dermatol. 1989;21(5 Pt 1):919-923.

Vakeva L, Sarna S, Vaalasti A, Pukkala E, Kariniemi AL, Rank A. A retrospective study of the probability of the evolution of parapsoriasis en plaques into mycosis fungoides. Acta Derm Venereol. 2005;85(4):318-323.

Sibbald C, Pope E. Systematic review of cases of cutaneous T-cell lymphoma transformation in pityriasis lichenoides and small plaque parapsoriasis. Br J Dermatol. 2016;175(4):807-809.

Cerroni L. Skin Lymphoma: The Illustrated Guide. 4th ed. Wiley Blackwell; 2014.

Jaffe ES, Arber DA, Campo E, Harris NL, Quintanilla-Martinez L. Hematopathology. 2nd ed. Elsevier; 2017.

Xavier JCC Jr, Ocanha-Xavier JP, Marques MEA. Shall we exclude parapsoriasis from the medical vocabulary? J Cutan Pathol. 2021;48(7):833–836.

Author Response

As the reviewer’s consideration, there is considerable debate on the terminology and relation of parapsoiasis to MF (Barnhill’s Dermatopathology, 4th ed. Pp 1184-1185). Parapsoriasis is broadly divided in two main types: small plaque parapsoriasis (SPP) and large plaque parapsoriasis (LPP). While SPP is generally considered a chronic benign condition, LPP is regarded as a premalignant dermatosis with a substantial risk of progression to mycosis fungoides (Weedon’s SkinPathology, 5th ed. Pp 114-115). Actually, 10% to 30% of large plaque parapsoriasis cases progresses to a frank cutaneous T-cell lymphoma (AP Lazar, WA Caro, HH Roenigk Jr, et al.: Parapsoriasis and mycosis fungoides: the Northwestern University experience, 1970 to 1985. J Am Acad Dermatol. 21:919-923 1989). Therefore, the major differential diagnosis of SPP and LPP includes early-stage MF and chronic eczema, which usually cannot be distinguished on histologic grounds alone (Barnhill’s Dermatopathology, 4th ed. Pp 1184-1185). In this study, after making pathologic diagnoses, dermatologists changed the clinical diagnosis from mycosis fungoides to parapsoriasis in 7 out of 22 non-MF cases, because they believe that there was still a possibility of the evolution of mycosis fungoides. We added the concerns and limitations of parapsoriasis cases classified as non-MF group in the Discussion section as below (page 8-9):

 In this study, after making pathologic diagnoses, dermatologists changed the clinical diagnosis from mycosis fungoides to parapsoriasis in 7 out of 22 non-MF cases. Parapsoriasis describes a group of cutaneous diseases that can be characterized by scaly patches or slightly elevated papules and/or plaques dispersed on the trunk or proximal extremities that have a resemblance to psoriasis [21]. Histologically, it shows cutaneous lymphoproliferations and refers to a heterogeneous group of uncommon dermatoses smilar to psoriasis. It is broadly divided in two main types: small plaque parapsoriasis (SPP) and large plaque parapsoriasis (LPP). While SPP is generally considered a chronic benign condition, LPP is regarded as a premalignant dermatosis with a substantial risk of progression to mycosis fungoides. Actually, 10% to 30% of LPP cases progresses to a frank cutaneous T-cell lymphoma [22]. It is known that monoclonal populations of T cells could be found in 20% or more of cases of LPP [23]. Therefore, the major differential diagnosis of parapsoriasis includes early-stage MF and chronic eczema, which usually cannot be distinguished on histologic grounds alone [24]. Thus, we included parapsoriasis as one of the mimickers of mycosis fungoides. However, there is considerable debate on the terminology and relation of parapsoiasis to MF. We believe that there was still a possibility of the evolution of mycosis fungoides in parapsoriasis cases which involved in this study, and parapsoriasis could not be entirely defined by the ISCL algorithm.

  1. Patterson, J.W. Weedon's Skin Pathology, 5 ed.; Elsevier: 2021; pp. 114-115.
  2. Lazar, A.P.; Caro, W.A.; Roenigk, H.H., Jr.; Pinski, K.S. Parapsoriasis and mycosis fungoides: the Northwestern University experience, 1970 to 1985. J Am Acad Dermatol 1989, 21, 919-923, doi:10.1016/s0190-9622(89)70277-2.
  3. Kikuchi, A.; Naka, W.; Harada, T.; Sakuraoka, K.; Harada, R.; Nishikawa, T. Parapsoriasis en plaques: its potential for progression to malignant lymphoma. J Am Acad Dermatol 1993, 29, 419-422, doi:10.1016/0190-9622(93)70204-7.
  4. Kempf, W.; Burg, G.; Massone, C.; Cerroni, L.; Pulitzer, M.; Crowson, A.N.; Magro, C.M. Barnhill's Dermatopathology, 4 ed.; Barnhill, R.L., Ed. Mc Graw Hill: 2020; pp. 1184-1185.

Reviewer 2 Report

I believe that the authors have mostly addressed my questions. I think that this paper is now acceptable for publication. 

Author Response

I would like to express the deepest gratitude for reviewer’s critical analysis of our manuscript and their helpful comments. We hope our responses are satisfactory.